# Contextual Markov Decision Processes using Generalized Linear Models

**Aditya Modi** [1]  **Ambuj Tewari** [2]

## Abstract

We consider the recently proposed reinforcement learning (RL) framework of Contextual Markov Decision Processes (CMDP), where the agent has a sequence of episodic interactions with tabular environments chosen from a possibly infinite set. The parameters of these environments depend on a context vector that is available to the agent at the start of each episode. In this paper, we propose a no-regret online RL algorithm in the setting where the MDP parameters are obtained from the context using generalized linear models (GLMs). The proposed algorithm `GL-ORL` relies on efficient online updates and is also memory efficient. Our analysis of the algorithm gives new results in the logit link case and improves previous bounds in the linear case. Our work is theoretical and we primarily focus on regret bounds but we aim to highlight the ubiquitous sequential decision making problem of learning *generalizable* policies for a population of individuals.

## 1. Introduction

Recent advances in reinforcement learning (RL) methods has led to increased focus on finding practical RL applications. RL algorithms provide a set of tools for tackling sequential decision making problems with potential applications ranging from web advertising and portfolio optimization, to healthcare applications like adaptive drug treatment. However, despite the empirical success of RL in simulated domains such boardgames and video games, it has seen limited use in real world applications because of the inherent trial-and-error nature of the paradigm. In addition to these concerns, for the applications listed above, the designer has to essentially design adaptive methods for a *population* of users instead of a single system. For example, consider the problem of optimizing adaptive drug treatment plans for a

sequence of patients. In such a scenario, one has to ensure quickly learning good policies for each user and also share the observed outcome data efficiently across patients. Intuitively, we expect that frequently seen patient types can be adequately dealt with by using adaptive learning methods whereas difficult and rare cases could be referred to experts.

In this paper, we consider a setting where at the start of every patient interaction, we have access to some *contextual information*. This information can be demographic, genomic or pertain to measurements taken from lab tests. We model this setting using the framework of Contextual Markov Decision Processes (CMDPs) previously studied by (Modi et al., 2018). Similar settings have been studied with slightly differing formalizations by (Abbasi-Yadkori & Neu, 2014; Hallak et al., 2015) and (Dann et al., 2018). While the framework proposed in these works is innovative, there are a number of deficiencies in the available set of results. First, theoretical guarantees (PAC-style mistake bounds or regret bounds) sometimes hold only under a restrictive linearity assumption on the mapping between contexts and MDPs. Second, if non-linear mappings are introduced, they do not always deal properly with the requirement that next-state distributions be properly normalized probability vectors.

We address these deficiencies using generalized linear models (GLMs) to model the mapping from context to MDP parameters. Our results provide new bounds for the logit link and improve existing results even in the simpler linear case. In addition, we focus on computational and space complexity concerns for learning in such CMDPs. Our proposed algorithm uses the popular optimism under the face of uncertainty (OFU) approach and relies on *efficient online updates*, both in terms of memory and computation time. It therefore differs from the typical OFU approaches whose running time scales linearly with number of observed contexts. Finally, the proposed algorithm also results in a cumulative policy certificate bound as studied by (Dann et al., 2018).

**Outline** In Section 2, we describe the setting formally with the required assumptions and notation. We present a method for confidence set construction in Section 3. Section 4 uses the confidence sets to develop our main algorithm `GL-ORL` and provides its theoretical analysis. In Section 6, we show a regret lower bound for our setting. After giving

---

[1]Computer Science and Engineering, Univ. of Michigan [2]Dept. of Statistics, Univ. of Michigan. Correspondence to: Aditya Modi <admodi@umich.edu>.

*Reinforcement Learning for Real Life (RL4RealLife) Workshop in the $36^{th}$ International Conference on Machine Learning*, Long Beach, California, USA, 2019. Copyright 2019 by the author(s).

a general method for obtaining confidence sets under prior information (Section 5) and a discussion of related work (Section 7), we conclude in Section 8.

## 2. Setting and Notation

The agent in this setting interacts with a sequence of fixed horizon tabular MDPs which are indexed by contexts $x_k \in \mathcal{X}$ where $\mathcal{X} \subset \mathbb{R}^d$. Each episode is a sequence of length $H : \{s_{k,1}, a_{k,1}, r_{k,1}, \ldots, s_{k,H}, a_{k,H}, r_{k,H}\}$ with states $s_{k,h} \in \mathcal{S}$, actions $a_{k,h} \in \mathcal{A}$ and rewards $r_{k,h} \in [0,1]$. In each episode, the state and reward for each timestep $h+1$ are sampled from the contextual MDPs ($M_k$) parameters: $P(\cdot|s_{k,h}, a_{k,h}; x_k)$ and $R(s_{k,h}, a_{k,h}; x_k)$. The actions $a_{k,h}$ are chosen by the agent's (contextual) policy $\pi_k$ computed at the beginning of each episode. We denote the size of MDP parameters by the usual notation: $|\mathcal{S}| = S$ and $|\mathcal{A}| = A$.

We denote the value of a policy in an episode $k$ by the expected total return for $H$ steps:

$$V_k^\pi = \mathbb{E}_{M_k, \pi_k} \left[ \sum_{h=1}^{H} r_{kh} \right]$$

The optimal policy for each episode $k$ is denoted by $\pi_k^* := \arg\max_\pi V_k^\pi$ and its value as $V^*$. We denote the *instantaneous regret* for episode $k$ as $\Delta_k = V_k^* - V_k^{\pi_k}$. The goal is to bound the *cumulative regret* $R(K)$, i.e., sum of these $\Delta_k$ for any number of episodes $K$. Note that this formulation allows an adversarial sequence of context vectors.

Additionally, for two matrices $X$ and $Y$, $\langle X, Y \rangle :=$ Trace($X^\top Y$). For a vector $x \in \mathbb{R}^d$ and a matrix $A \in \mathbb{R}^{d \times d}$, we define $\|x\|_A^2 := x^\top A x$. For a matrices $W \in \mathbb{R}^{m \times n}$ and $X \in \mathbb{R}^{n \times n}$, we have $\|W\|_X^2 := \sum_{i=1}^{m} \|W^{(i)}\|_X^2$ where $W^{(i)}$ is the $i^{\text{th}}$ row of the matrix. For simplicity, we remove the subscripts/superscripts from the notation when clear from the context. Any norm which appears without a subscript will denote the $\ell_2$ norm.

### 2.1. Generalized Linear Model for CMDPs

We assume that each contextual MDP $M_k$ is obtained by a set of generalized linear models. Specifically, for each pair $s, a \in \mathcal{S} \times \mathcal{A}$, there exists a weight matrix $W_{sa}^* \in \mathcal{W} \subseteq \mathbb{R}^{S \times d}$ where $\mathcal{W}$ is a convex set[1]. For any context $x_k \in \mathbb{R}^d$, the next state distribution for the pair is specified by a GLM:

$$P_k(\cdot|s,a) = \nabla\Phi(W_{sa}^* x_k) \tag{1}$$

where $\Phi(\cdot) : \mathbb{R}^S \to \mathbb{R}$ is the link function of the GLM. We will assume that this link function is convex which is always the case for a canonical exponential family (Lauritzen, 1996). For rewards, we assume that each mean reward is

---

[1] Without loss of generality, we can set the last row $W_S^*$ of the weight matrix to be 0 to avoid an overparametrized system.

given by a linear function of the context: $r_k(s,a) := \theta_{sa}^{*\top} x_k$ where $\theta^* \in \Theta \subseteq \mathbb{R}^d$. In addition, we will make the following assumptions about the link function.

**Assumption 2.1.** *The function $\Phi(\cdot)$ is $\alpha$-strongly convex:*

$$\Phi(v) \geq \Phi(u) + \langle \nabla\Phi(u), v - u \rangle + \tfrac{\alpha}{2}\|u - v\|_2^2 \tag{2}$$

**Assumption 2.2.** *The function $\Phi(\cdot)$ is $L$-strongly smooth:*

$$\Phi(v) \leq \Phi(u) + \langle \nabla\Phi(u), v - u \rangle + \tfrac{\beta}{2}\|u - v\|_2^2 \tag{3}$$

We will see that this assumption is critical for constructing the confidence sets used in our algorithm. We make another assumption about the size of the weight matrices $W_{sa}^*$ and contexts $x_k$:

**Assumption 2.3.** *For all episodes $k$, we have $\|x_k\|_2 \leq R$ and for all state-action pairs $(s, a)$, $\|W_{sa}^{*(i)}\|_2 \leq B_p$ and $\|\theta^*\|_2 \leq B_r$. So, we have $\|Wx\|_\infty \leq B_p R$ for all $W \in \mathcal{W}$.*

The regret bounds for the proposed algorithms will depend on these quantities.

### 2.2. Special Cases

We show that this setting covers contextual MDPs constructed through multinomial logit models or linear combination of base MDPs.

**Example 2.4** (Multinomial logit model, (Agarwal, 2013)). *Each next state is sampled from a multinomial distribution with probabilities:*

$$p(s_i|s,a) = \frac{\exp(W_{sa}^{(i)} x)}{\sum_{j=1}^{S} \exp(W_{sa}^{(j)} x)}$$

*The link function for this case can be given as $\Phi(y) = \log(\sum_{i=1}^{S} \exp(y_i))$ which can be shown to be strongly convex with $\alpha = \frac{1}{\exp(BR)S^2}$ and smooth with $\beta = 1$.*

**Example 2.5** (Linear combination of MDPs, (Modi et al., 2018)). *Each MDP is obtained by a linear combination of $d$ base MDPs. Here, $x_k \in \Delta_{d-1}$, and $P_k(\cdot|s,a) := \sum_{i=1}^{d} x_{ki} P^i(\cdot|s,a)$. The link function for this can be shown to be:*

$$\Phi(y) = \tfrac{1}{2}\|y\|_2^2$$

*which is strongly convex and smooth with parameters $\alpha = \beta = 1$. Moreover, $W^*$ here is the $S \times d$ matrix containing each next state distribution in a column. We have, $B_p \leq \sqrt{d}$, $\|W^*\|_F \leq \sqrt{d}$ and $\|W^* x_k\|_2 \leq 1$.*

## 3. Online Estimates and Confidence Set Construction

In order to obtain a no-regret algorithm for our setting, we will follow the popular optimism under uncertainty approach

which relies of the construction of confidence sets for MDP parameters at the beginning of each episode. We focus on deriving these confidence sets for the next state distributions for all state action pairs. In this section, we will assume that, for all pairs $(s, a)$, $\Phi$, $\alpha$, $B$ and $R$ are known apriori. Therefore, for each state-action pair, we have the following online estimation problem: For $t = 1, 2, \ldots$:

- Propose an estimate $W_t$ and a set $\mathcal{C}$ such that, $W^* \in \mathcal{C}$ with high probability.

- Observe $x_t \in \mathcal{X}$ and a sample $y_t \sim \nabla\Phi_t(W^*x_t)$ where $y_t$ denotes the one-hot vector with the value 1 at the sampled state $s_t$.

We consider this as an online optimization problem with the following loss sequence based on the negative log-likelihood:

$$l_t(W; x_t, y_t) = \Phi(Wx_t) - y_t^\top Wx_t \qquad (4)$$

The utility of this loss function is that it preserves strong convexity of $\Phi$ with respect to $Wx_t$ and is a proper loss function (Agarwal, 2013):

$$\arg\min_W \mathbb{E}\big[l_t(W; x_t, y_t)|x_t\big] = W^* \qquad (5)$$

Since our aim is computational and memory efficiency, we carefully follow the Online Newton Step (Hazan et al., 2007) based method proposed for 0/1 rewards with logistic link function in (Zhang et al., 2016). This extension to generalized linear models utilizes the structure of multinomial vectors in various places of the analysis. Let us focus on the estimation problem for a single state-action pair. The update rule for the parameter matrix $W_t$ is given below.

Let $W_1 = \mathbf{0}$ and $Z_1 = \lambda\mathbb{I}_d$. We maintain an estimate $W_{t+1}$ by the following update rule:

$$\arg\min_{W \in \mathcal{W}} \frac{\|W - W_t\|_{Z_{t+1}}^2}{2} + \eta\langle\nabla l_t(W_tx_t)x_t^\top, W - W_t\rangle \qquad (6)$$

where $Z_{t+1} = Z_t + \frac{\eta\alpha}{2}x_tx_t^\top$.

We will see that strong convexity of $\Phi(\cdot)$ plays an important role in the analysis and the multinomial GLM assumption improves the dependence on $S$. The method only stores the empirical covariance matrix and solves an optimization problem using only the current context. Since, the set $\mathcal{W}$ is convex, this is a tractable problem and can be solved via any off-the shelf optimizer up to desired accuracy. The computation time for computing these sets for each context is $O(\text{poly}(S, A, d))$ with no dependence on $t$. Furthermore, we only store $SA$-many matrices of size $S \times d$ and covariance matrices of sizes $d \times d$. In Section 4, we will see that we can obtain an $\ell_1$ confidence set over the next state

distribution for each context $x_t$ if we have access to a confidence set for $W^*$. The following key result gives such a confidence set.

**Theorem 3.1** (Confidence set for $W^*$). *If $W_t$ is obtained by equation 6 for all timesteps $t = 1, 2, \ldots$, then for all timesteps, with probability at least $1 - \delta$, we have:*

$$\|W_{t+1} - W^*\|_{Z_{t+1}} \leq \sqrt{\gamma_{t+1}} \qquad (7)$$

*with*

$$\begin{aligned}\gamma_{t+1} &= \lambda\|W^*\|_F^2 + 8\eta BR \\ &\quad + 2\eta\Big[\big(\tfrac{4}{\alpha} + \tfrac{8}{3}BR\big)\tau_t + \tfrac{4}{\alpha}\log\tfrac{\det(Z_{t+1})}{\det(Z_1)}\Big]\end{aligned} \qquad (8)$$

*with $\tau_t = \log(2\lceil 2\log t\rceil t^2/\delta)$.*

The term $\gamma_t$ now depends on the size of the true weight matrix, strong convexity parameter $\frac{1}{\alpha}$ and the log determinant of the covariance matrix. We will see later that the term is of the order $\mathcal{O}(d\log t)$. Therefore, overall this term has scales as $\mathcal{O}(S + \frac{d}{\alpha}\log^2 t)$.

### 3.1. Proof of Theorem 3.1

We closely follow the analysis from (Zhang et al., 2016) and use properties of the multinomial output to adapt it to our case. We will denote the derivative with respect to the matrix for loss $l_t$ as $\nabla l_t(W_t)$ and the derivative with respect to the projection as $\nabla l_t(W_tx_t)$. Now, using the strong convexity of the loss function $l_t$, for all $t$, we have:

$$\begin{aligned}l_t(W_t) - l_t(W^*) &\leq \langle\nabla l_t(W_tx_t), W_tx_t - W^*x_t\rangle \\ &\quad - \frac{\alpha}{2}\underbrace{\|W^*x_t - W_tx_t\|_2^2}_{:=b_t}\end{aligned}$$

Taking expectation w.r.t. the multinomial sample $y_t$, we get:

$$\begin{aligned}&\mathbb{E}_{y_t}[l_t(W_t) - l_t(W^*)] \\ &\leq \mathbb{E}_{y_t}[\langle\nabla l_t(W_tx_t), W_tx_t - W^*x_t\rangle] - \tfrac{\alpha}{2}b_t \\ 0 &\leq \mathbb{E}_{y_t}[\langle\nabla l_t(W_tx_t), W_tx_t - W^*x_t\rangle] - \tfrac{\alpha}{2}b_t \quad (9)\end{aligned}$$

where the lhs is obtained by using the calibration property from eq. 5. Now, for the first term on rhs, we have:

$$\begin{aligned}&\mathbb{E}_{y_t}[\langle\nabla l_t(W_tx_t), W_tx_t - W^*x_t\rangle] \\ &= \mathbb{E}_{y_t}[\langle\nabla\Phi(W_tx_t) - y_t, W_tx_t - W^*x_t\rangle] \\ &= (\tilde{p}_t - p_t)^\top(W_t - W^*)x_t \\ &= \underbrace{(\tilde{p}_t - y_t)^\top(W_t - W^*)x_t}_{:=\mathbf{I}} \\ &\quad + \underbrace{(y_t - p_t)^\top(W_t - W^*)x_t}_{:=c_t} \qquad (10)\end{aligned}$$

where $\tilde{p}_t = \nabla\Phi(W_tx_t)$ and $\mathbb{E}[y_t] = p_t = \nabla\Phi(W^*x_t)$. We bound the term $\mathbf{I}$ using the following lemma:

**Lemma 3.2.**

$$\langle \nabla l_t(W_t x_t), W_t x_t - W^* x_t \rangle$$
$$\leq \quad \frac{\|W_t - W^*\|_{Z_{t+1}}}{2\eta} - \frac{\|W_{t+1} - W^*\|_{Z_{t+1}}}{2\eta}$$
$$+ 2\eta \|x_t\|^2_{Z_{t+1}^{-1}} \tag{11}$$

*Proof.* To prove this, we go back to the update rule in (6) which has the following form:

$$Y = \underset{W \in \mathcal{W}}{\arg\min} \frac{\|W - X\|^2_M}{2} + \eta a^\top W b$$

with $Y = W_{t+1}$, $X = W_t$, $a = \nabla l_t(W_t x_t) = \tilde{p}_t - y_t$, $b = x_t$ and $M = Z_{t+1}$. For a solution to any such optimization problem, by the first order optimality conditions, we have:

$$\langle (Y - X)M + \eta a b^\top, W - Y \rangle \geq 0$$
$$(Y - X)MW \geq (Y - X)MY$$
$$- \eta a^\top (W - Y)b$$

Now,

$$\|X - W\|^2_M - \|Y - W\|^2_M$$
$$= \sum_{i=1}^S X^i M X^i + W^i M W^i - Y^i M Y^i$$
$$- W^i M W^i + 2(Y^i - X^i)MW^i$$
$$\geq \|X - Y\|^2_M - 2\eta a^\top (W - Y)b$$
$$= \|X - Y\|^2_M + 2\eta a^\top (Y - X)b$$
$$- 2\eta a^\top (W - X)b$$
$$\geq \underset{A \in \mathbb{R}^{S \times d}}{\arg\min} \|A\|^2_M + 2\eta a^\top A b$$
$$- 2\eta a^\top (W - X)b \tag{12}$$

Noting that $a = \tilde{p}_t - y^t$, we get

$$\underset{A \in \mathbb{R}^{S \times d}}{\arg\min} \|A\|^2_M + 2\eta a^\top A b \geq \sum_{i=1}^S -\eta^2 a_i^2 \|b\|^2_{M^{-1}}$$
$$\geq -4\eta^2 \|b\|^2_{M^{-1}}$$

Substituting this and $W = W^*$ along with other terms in ineq. (12) proves the stated lemma (ineq. (11)). $\square$

For bounding $c_t$ in ineq. 10, we note that $c_t$ is a martingale difference sequence for which we have:

$$|c_t| = (y_t - p_t)^\top (W_t - W^*)x_t$$
$$\leq \|(y_t - p_t)\|_1 \|(W_t - W^*)x_t\|_\infty$$
$$\leq 4BR$$

Similarly, for martingale $C_t := \sum_{i=1}^t c_i$, we bound the conditional variance as

$$\Sigma_t^2 = \sum_{i=1}^t \mathbb{E}_{y_i} \left[ \left( (y_t - p_t)^\top (W_t - W^*)x_t \right)^2 \right]$$
$$\leq \sum_{i=1}^t \mathbb{E}_{y_i} \left[ \left( y_t^\top (W_t - W^*)x_t \right)^2 \right]$$
$$\leq \sum_{i=1}^t 4B^2 R^2$$

Now, using Bernstein's inequality for martingales and the peeling technique as used in Lemma 5 of (Zhang et al., 2016), we get

$$\sum_{i=1}^t c_i \leq 4BR + 2\sqrt{\tau_t \sum_{i=1}^t b_i} + \tfrac{8}{3} BR \tau_t$$

with probability at least $1 - \delta$. Using the RMS-AM inequality, with probability at least $1 - \delta$, we get

$$\sum_{i=1}^t c_i \leq 4BR + \frac{\alpha}{4} \sum_{i=1}^t b_i + \left( \frac{4}{\alpha} + \frac{8}{3} BR \right) \tau_t \tag{13}$$

From eqs. (9), (10) and (11), we have

$$\|W_{t+1} - W^*\|_{Z_{t+1}}$$
$$\leq \|W_t - W^*\|_{Z_t} - \frac{\eta\alpha}{2} b_t + 2\eta c_t + 4\eta^2 \|x_t\|^2_{Z_{t+1}^{-1}}$$

Unwrapping the `rhs` over $t$ and substituting ineq. (13), we get

$$\|W_{t+1} - W^*\|_{Z_{t+1}}$$
$$\leq \|W^*\|_{Z_1} + 2\eta \left[ 4BR + \left( \frac{4}{\alpha} + \frac{8}{3} BR \right) \tau_t \right]$$
$$+ 4\eta^2 \sum_{i=1}^t \|x_t\|^2_{Z_{t+1}^{-1}}$$
$$\leq \lambda \|W^*\|_F^2 + 2\eta \left[ 4BR + \left( \frac{4}{\alpha} + \frac{8}{3} BR \right) \tau_t \right]$$
$$+ 4\eta^2 \sum_{i=1}^t \|x_t\|^2_{Z_{t+1}^{-1}}$$

Using the following Lemma from (Zhang et al., 2016), we arrive at the result in Theorem 3.1.

**Lemma 3.3** (Lemma 6, (Zhang et al., 2016))*. We have,*

$$\sum_{i=1}^t \|x_t\|^2_{Z_{t+1}^{-1}} \leq \frac{2}{\eta\alpha} \log \frac{\det(Z_{t+1})}{\det(Z_1)}$$

# 4. Optimistic Reinforcement Learning for GLM CMDP

In this section, we describe the OFU based online learning algorithm which leverages the confidence sets as described in the previous sections. Not surprisingly, our algorithm is similar to the algorithm of (Dann et al., 2018) and (Abbasi-Yadkori & Neu, 2014) and follows the standard format for no-regret bounds in MDPs. As can be seen from the algorithm outline, we construct confidence sets for the next state distributions and rewards for each state-action pair $(s, a)$. Using that, we compute an optimistic policy to run at the beginning of each episode. We will use $x_k$ to denote the context for episode $k$ and the online update in Section 3 works with a total of $T = KH$ steps. Now, we can state the confidence interval for $p_k(\cdot|s, a)$ as:

$$
\begin{aligned}
\xi_{k,sa}^{(p)} &:= \|p_k(\cdot|s, a) - \hat{p}_k(\cdot|s, a)\|_1 \\
&\leq \beta\sqrt{S}\|W_{sa}^* - \overline{W}_{k,sa}\|_{Z_{k,sa}}\|x_k\|_{Z_{k,sa}^{-1}} \\
&\leq \beta\sqrt{S}\sqrt{\gamma_{k,sa}}\|x_k\|_{Z_{k,sa}^{-1}} \quad (14)
\end{aligned}
$$

where quantities with subscript $k$ denote the value at the beginning of episode $k$. The pair of matrices $\overline{W}, Z$ is maintained for every state action pair $(s, a)$. For rewards, we assume the usual linear bandit structure, which has a standard confidence interval ((Lattimore & Szepesvári, 2018), Theorem 20.5).

$$
\xi_{k,sa}^{(r)} = \underbrace{\left(\sqrt{\lambda d} + \sqrt{\tfrac{1}{4}\log\frac{\det Z_{k,sa}}{\delta_r^2 \det \lambda I}}\right)}_{:=\zeta_{k,sa}} \|x_k\|_{Z_{k,sa}^{-1}} \quad (15)
$$

For the given algorithm using the confidence sets from previous sections, we can show the following bound:

**Theorem 4.1** (Regret of GL-ORL). *For any $\delta \in (0, 1)$, if Algorithm 4 is run with the estimators from Section 3, then for all $K \in \mathbb{N}$ and with probability at least $1 - \delta$, the regret $R(K)$ is:*

$$
\tilde{\mathcal{O}}\left(\left(\frac{\sqrt{d}\max_{s,a}\|W_{sa}^*\|_F}{\sqrt{\alpha}} + \frac{d}{\alpha}\right)\beta S H^2 \sqrt{AK}\log\frac{KHd}{\lambda\delta}\right)
$$

## 4.1. Proof of Theorem 4.1

We first begin by showing that the computed policy's value is optimistic.

**Lemma 4.2** (Optimism). *If all the confidence intervals as computed in Algorithm 4 are valid for all episodes $k$, then for all $k$ and $h \in [H]$ and $s, a \in \mathcal{S} \times \mathcal{A}$, we have:*

$$
\tilde{Q}_{k,h}(s, a) \geq Q_{k,h}^*(s, a)
$$

*Proof.* For every episode, the lemma is true trivially for $H + 1$. Assume that it is true for $h + 1$. Now, for $h$, we

---

**Algorithm 1** GL-ORL (Generalized Linear Optimistic Reinforcement Learning)

**Input:** $\mathcal{S}, \mathcal{A}, H, \Phi, d, \mathcal{W}, \lambda, \delta$
$\delta' = \frac{\delta}{2SA+SH}$, $\tilde{V}_{k,H+1}(s) = 0 \; \forall s \in \mathcal{S}, k \in \mathbb{N}$
**for** $k \leftarrow 1, 2, 3, \dots$ **do**
  Observe current context $x_k$
  **for** $s \in \mathcal{S}, a \in \mathcal{A}$ **do**
    $\hat{p}_k(\cdot|s, a) \leftarrow \nabla\Phi(\hat{W}_{k,sa}x_k)$
    $\hat{r}_k(s, a) \leftarrow \langle\hat{\theta}_{k,sa}, x_k\rangle$
    Compute confidence intervals using eqns. (14), (15)
  **end for**
  **for** $h \leftarrow H, H-1, \cdots, 1$, and $s \in \mathcal{S}$ **do**
    **for** $a \in \mathcal{A}$ **do**
      $\varphi(s, a) = \|\tilde{V}_{k,h+1}\|_\infty \xi_{k,sa}^{(p)} + \xi_{k,sa}^{(r)}$
      $\tilde{Q}_{k,h}(s, a) = 0 \vee (\hat{p}_k(\cdot|s, a)^\top \tilde{V}_{k,h+1} + \hat{r}_k(s, a) + \varphi(s, a)) \wedge V_h^{\max}$
    **end for**
    $\pi_{k,h}(s) = \arg\max_a \tilde{Q}_{k,h}(s, a)$
    $\tilde{V}_{k,h}(s) = \tilde{Q}_{k,h}(s, \pi_{k,h}(s))$
  **end for**
  **Sample episode** using policy $\pi_k$
**end for**

---

have:

$$
\begin{aligned}
&\tilde{Q}_{k,h}(s, a) - Q_{k,h}^*(s, a) \\
&= (\hat{p}_k(s, a)^\top \tilde{V}_{k,h+1} + \hat{r}_k(s, a) + \varphi(s, a)) \wedge V_h^{\max} \\
&\quad - p_k(s, a)^\top V_{k,h+1}^* - r_k(s, a) \\
&= \hat{r}_k(s, a) - r_k(s, a) + \hat{p}_k(s, a)^\top (\tilde{V}_{k,h+1} - V_{k,h+1}^*) \\
&\quad + \varphi(s, a) - (p_k(s, a) - \hat{p}_k(s, a))^\top V_{k,h+1}^* \\
&\geq -|\hat{r}_k(s, a) - r_k(s, a)| + \varphi \\
&\quad - \|p_k(s, a) - \hat{p}_k(s, a)\|_1 \|\tilde{V}_{k,h+1}\|_\infty \\
&\geq 0
\end{aligned}
$$

where the last line uses the guarantee on confidence intervals and the assumption for $h + 1$. $\qquad\square$

Using the optimism guarantee, we have

$$
\begin{aligned}
\Delta_k &\leq \tilde{V}_{k,1}(s) - V_{k,1}^{\pi_k}(s) \\
&\leq (\hat{p}_k(s, a)^\top \tilde{V}_{k,2} + \hat{r}_k(s, a) + \varphi) \wedge V_1^{\max} \\
&\quad - p_k(s, a)^\top V_{k,2}^{\pi_k} - r_k(s, a) \\
&\leq (\varphi + \hat{p}_k(s, a) - p_k(s, a))^\top \tilde{V}_{k,2} + \hat{r}_k(s, a) \\
&\quad - r_k(s, a)) \wedge V_1^{\max} + p_k(s, a)^\top (V_{k,2}^{\pi_k} - \tilde{V}_{k,2}) \\
&\leq 2\varphi \wedge V_1^{\max} + p_k(s, a)^\top (V_{k,2}^{\pi_k} - \tilde{V}_{k,2}) \\
&\leq \sum_{h,s,a} \left[ \mathbb{P}_k[s_h, a_h = s, a | s_{k,1}] \right. \\
&\qquad\qquad \left. (2\varphi(s, a) \wedge V_h^{\max}) \right] \quad (16)
\end{aligned}
$$

Therefore, we have:

$$R(K) := \sum_{k=1}^{K} \Delta_k$$

$$\leq \sum_{k=1}^{K}\sum_{h=1}^{H}\sum_{s,a} \Big( \mathbb{P}_k[s_{k,h}=s, a_{k,h}=a|s_{k,1}]$$

$$- \mathbb{I}[s_{k,h}=s, a_{k,h}=a]\Big)V_h^{\max}$$

$$+ \sum_{k=1}^{K}\sum_{h=1}^{H}\mathbb{I}[s_{k,h}=s, a_{k,h}=a]$$

$$\cdot (2\varphi(s_{k,h}, a_{k,h}) \wedge V_h^{\max})$$

We bound the first summation using Lemma 23 of (Dann et al., 2018).

**Lemma 4.3** ((Dann et al., 2018), Lemma 23). *With probability at least $1 - SH\delta_1$, for all $K \in \mathbb{N}$, we have*

$$\sum_{k=1}^{K}\sum_{h=1}^{H}\sum_{s,a} \Big( \mathbb{P}_k[s_{k,h}=s, a_{k,h}=a|s_{k,1}]$$

$$- \mathbb{I}[s_{k,h}=s, a_{k,h}=a] \Big) \leq SH\sqrt{K \log \frac{6\log(2K)}{\delta_1}}$$

Now, we bound the second term of the regret bound decomposition:

$$\sum_{k=1}^{K}\sum_{h=1}^{H}\mathbb{I}_{k,h}(s,a)(2\varphi(s_{k,h}, a_{k,h}) \wedge V_h^{\max})$$

$$\leq \sum_{k=1}^{K}\sum_{h=1}^{H}\mathbb{I}_{k,h}(s,a)(2\xi_{k,s_{k,h},a_{k,h}}^{(r)} \wedge V_h^{\max})$$

$$+ \sum_{k=1}^{K}\sum_{h=1}^{H}\mathbb{I}_{k,h}(s,a)(2V_{h+1}^{\max}\xi_{k,s_{k,h},a_{k,h}}^{(p)} \wedge V_h^{\max})$$

Now, we can bound the individual terms, where we focus on the higher order terms arising due to $\xi^{(p)}$:

$$\sum_{k=1}^{K}\sum_{h=1}^{H}(2V_{h+1}^{\max}\xi_{k,s_{k,h},a_{k,h}}^{(p)} \wedge V_h^{\max})$$

$$\leq 2\sum_{k=1}^{K}\sum_{h=1}^{H}V_h^{\max}\Big(1 \wedge \beta\sqrt{S\gamma_k(s_{k,h}, a_{k,h})}\|x_k\|_{Z_{k,sa,h}^{-1}}\Big)$$

$$\leq 2\beta V_1^{\max}\sqrt{\frac{2S\gamma_T}{\eta\alpha}}\sum_{k=1}^{K}\sum_{h=1}^{H}\Big(1 \wedge \sqrt{\frac{\eta\alpha}{2}}\|x_k\|_{Z_{k,sa,h}^{-1}}\Big)$$

$$\leq 2\beta V_1^{\max}\sqrt{\frac{2S\gamma_T T}{\eta\alpha}}\sqrt{\sum_{k=1}^{K}\sum_{h=1}^{H}\Big(1 \wedge \frac{\eta\alpha}{2}\|x_k\|_{Z_{k,sa,h}^{-1}}^2\Big)}$$

where the last inequality follows from Cauchy-Schwartz inequality. We now bound the elliptic potential inside the

square root. Note that, instead of summing up the weighted operator norm with changing values of $Z_t$, we keep the matrix same for all observations in an episode. Note that, $Z_k$ denotes the matrix at the beginning of episode $k$ and therefore, does not include the terms $x_k x_k^\top$. Therefore, we re-derive the bound for this setting. Now, for any episode $k$:

$$\sum_{h=1}^{H}\Big(1 \wedge \frac{\eta\alpha}{2}\|x_k\|_{Z_{k,sa,h}^{-1}}^2\Big)$$

$$\leq 2\sum_{s,a}\sum_{h=1}^{H}\mathbb{I}_{k,h}(s,a)\log\Big(1 + \frac{\eta\alpha}{2}\|x_k\|_{Z_{k,sa}^{-1}}^2\Big)$$

$$\leq 2\sum_{s,a}N_k(s,a)\log\Big(1 + N_k(s,a)\frac{\eta\alpha}{2}\|x_k\|_{Z_{k,sa}^{-1}}^2\Big)$$

$$= 2H\sum_{s,a}\log\Big(\frac{\det Z_{k+1,sa}}{\det Z_{k,sa}}\Big)$$

where in the last step, we have used the following:

$$Z_{k+1} = Z_k^{1/2}(1 + \frac{\eta\alpha}{2}N_k Z_k^{-1/2}x_k x_k^\top Z_k^{-1/2})Z_k^{1/2}$$

$$\det Z_{k+1} = \det Z_k\Big(1 + N_k\frac{\eta\alpha}{2}\|x_k\|_{Z_k^{-1}}^2\Big)$$

Therefore, we can finally bound the term as:

$$\sum_{k=1}^{K}\sum_{h=1}^{H}(2V_{h+1}^{\max}\xi_{k,s_{k,h},a_{k,h}}^{(p)} \wedge V_h^{\max})$$

$$\leq 4\beta V_1^{\max}\sqrt{\frac{2S\gamma_T T}{\eta\alpha}}\sqrt{2H\sum_{s,a}\log\frac{\det Z_{K+1}}{\det \lambda I}}$$

$$= 4\beta V_1^{\max}\sqrt{\frac{2S\gamma_T T}{\eta\alpha}}\sqrt{2HSAd\log\Big(1 + \frac{TR^2}{\lambda d}\Big)}$$

The regret component for rewards will contain lower order terms and we ignore it for conciseness. Now, taking $\delta_1 = \delta_p = \delta_r = \delta/(2SA + SH)$, we get the total failure probability for Lemma 4.3 and the confidence intervals to be at most $\delta$. If we use the results from Section 3, we know that $\gamma_T = \tilde{\mathcal{O}}\Big(\|W^*\|_F^2 + \frac{d}{\alpha}\log\Big(1 + \frac{TR^2}{\lambda d}\Big)\Big)$. Thus, by combining all terms, we get

$$R(K) = \tilde{\mathcal{O}}\Big(\Big(\frac{\sqrt{d}\max_{s,a}\|W_{sa}^*\|_F}{\sqrt{\alpha}} + \frac{d}{\alpha}\Big)\beta SH^2\sqrt{AK}\Big)$$

Substituting the bounds on $\|W_{sa}^*\|_F^2$, we have the following two corollaries.

**Corollary 4.4** (Multinomial logit model). *For example 2.4, we have $\|W^*\|_F \leq B\sqrt{S}$, $\alpha = \frac{1}{\exp(BR)S^2}$ and $\beta = 1$. Therefore, the regret bound of Algorithm 4 is $\tilde{\mathcal{O}}(dS^3H^2\sqrt{AK})$.*

**Corollary 4.5** (Regret bound for linear combination case). *For example 2.5, with $\|W^*\|_F \leq \sqrt{d}$, the regret bound of Algorithm 4 is $\tilde{\mathcal{O}}(dSH^2\sqrt{AK})$.*

For the linear case, as considered by (Dann et al., 2018), our guarantee improves by a factor of $\tilde{\mathcal{O}}(\sqrt{S})$ which can be attributed to our use of improved estimators for next-state distribution for each state-action pair. For GLMs, the guarantee provided by (Abbasi-Yadkori & Neu, 2014) is not comparable with our case as their assumptions considered invalid MDPs. Specifically, their setting assumes that the probability distribution for each next state is a GLM which does not create normalized next state-distributions. However, even if we ignore their modelling error and the strong convexity coefficient in our result, we still get an $\tilde{\mathcal{O}}(S\sqrt{AH})$ improvement. Further, as the confidence intervals in Section 3 are two-sided, we would also get a policy certificate bound of the same order with the same improvement over (Dann et al., 2018). In the next section, we show a lower bound for both the linear and logit cases. Our lower bound highlights a gap of $\tilde{\mathcal{O}}(H\sqrt{dS})$ and $\tilde{\mathcal{O}}(S^2H\sqrt{dS})$ for the two cases respectively.

### 4.2. Mistake bound for the setting

In their paper, (Dann et al., 2018) claim that showing a PAC style mistake bound for this setting may require an entirely novel confidence set construction. The key issue is the increasing size of these sets due to a factor of $\log t$. However, it is still important for practical purposes to have an understanding of how the number of mistakes scale with the total number of episodes. For example, in healthcare, this will be akin to bounding the error rate of treatments. Here, a mistake is defined as an episode in which the value of the learner's policy $\pi_k$ is not $\epsilon$-optimal, i.e., $V_k^* - V_k^{\pi_k} \geq \epsilon$. In our setting, we can show the following result.

**Theorem 4.6** (Bound on the number of mistakes). *For any number of episodes $K$, $\delta \in (0,1)$ and $\epsilon \in (0,H)$, with probability at least $1 - \delta$, the number of episodes where* GL-ORL's *policy $\pi_k$ is not $\epsilon$-optimal is bounded by*

$$\mathcal{O}\left(\frac{dS^2AH^5\log(KH)}{\epsilon^2}\left(\frac{d\log^2(KH)}{\alpha} + S\right)\right)$$

*ignoring $\mathcal{O}(poly(\log\log KH))$ terms.*

We defer the proof to Appendix A. Note that this term depends poly-logarithmically on $K$ and therefore increases with time. The algorithm doesn't need to know the value of $\epsilon$ and gives this guarantee for all $\epsilon$. In contrast, PAC algorithms typically use the $\epsilon$ value to be compute the optimistic bonus. Also, guaranteeing that a certain episode will not incur a mistake requires a tight confidence set or a oracle which guarantees the required threshold. As such, we can either obtain such an algorithm by modifying the confidence sets or combining the algorithm with a KWIK oracle (Li et al., 2011). Therefore, we expect that such a bound can be obtained and leave it for future work.

## 5. Improved Confidence Sets for Structured Spaces

In Section 3, we derived confidence sets for $W^*$ for the case when it lies in a bounded set. However, in many cases, we have additional prior knowledge about the problem in terms of possible constraints over the set $\mathcal{W}$. For example, consider the healthcare inspired scenario where the context vector is the genomic encoding of the patient. For treating a given disease, it is fair to assume that the disease-progression of the patient depends on a few genes rather than the entire genome which suggests a sparse dependence of the transition model on the context vector $x$. In terms of the parameter $W^*$, this translates as complete columns of the matrix being zeroed out for the irrelevant indices. Thus, we may want to consider methods for constructing confidence sets which take this specific structure into account.

In this section, we show that it is possible to convert a generic regret guarantee of an online learner to a confidence set. If the online learner adapts to the structure of $\mathcal{W}$, we would get the aforementioned improvement. Our conversion proof presented here is reminiscent of the techniques used in (Abbasi-Yadkori et al., 2012) and (Jun et al., 2017) with close resemblance to the latter. For this section, we use $X_t$ to denote the $t \times d$ shaped matrix with each row as $x_i$ and $C_t$ as $t \times S$ shaped matrix with each row $i$ being $(W_ix_i)^\top$. Also, set $\overline{W}_t := Z_{t+1}^{-1}X_t^\top C_t$. Using a similar notation as before, we can give the following guarantee.

**Theorem 5.1** (Multinomial GLM Online-to-confidence set conversion). *Suppose losses $l_i$ are $\alpha$-strongly convex. If there exists an online learning oracle which takes in the sequence $\{x_i, y_i\}_{i=1}^t$, and produces outputs $\{W_i\}_{i=1}^t$ with bounded regret for all $W \in \mathcal{W}$ and $t \geq 1$:*

$$\sum_{i=1}^{t} l_i(W_i) - l_i(W) \leq B_t,$$

*then with the centers $\overline{W}_t$ defined above, we have, with probability at least $1 - \delta$, for all $t \geq 1$, we have*

$$\|W^* - \overline{W}_t\|_{Z_{t+1}}^2 \leq \gamma_t,$$

*where $\gamma_t := \gamma_t'(B_t) + \lambda B^2 S - (\|C_t\|_F^2 - \langle\overline{W}_t, X_t^\top C_t\rangle)$ with*

$$\gamma_t'(B_t) := 1 + \frac{4}{\alpha}B_t + \frac{8}{\alpha^2}\log\left(\frac{1}{\delta}\sqrt{4 + \frac{8B_t}{\alpha} + \frac{16}{\alpha^4\delta^2}}\right).$$

We defer the proof to Appendix C due to space constraints. Note that, all quantities required in the expression $\gamma_t$ can be incrementally computed. The required quantities are $Z_t$ and $Z_t^{-1}$ along with $X_t^\top C_t$ which are incrementally updated with $O(poly(S, d))$ computation. Also, we note that this confidence set is meaningful when $B_t$ is poly-logarithmic

in $t$ which is possible for strongly convex losses as shown in (Jun et al., 2017). The dependence on $S$ and $d$ is the same as the previous construction, but the dependence on the strong convexity parameter is worse.

**Column sparsity of $W^*$** Similar to sparse stochastic linear bandit, as discussed in (Abbasi-Yadkori et al., 2012), one can use an online learning method with the group norm regularizer ($\|W\|_{2,1}$). Therefore, if there exists an online no-regret algorithm which is efficient and has improved dependence on the sparsity coefficient $p$, we can get an $O(\sqrt{p \log d})$ size confidence set improving the final regret bound to $\tilde{\mathcal{O}}(\sqrt{pdT})$ as observed in the linear bandit case. To our knowledge, even in the sparse adversarial linear regression setting, obtaining an efficient and sparsity aware regret bound is an open problem. The algorithms proposed by (Gaillard & Wintenberger, 2018) can potentially lead to such an online learning algorithm and we leave this for future work.

## 6. Lower Bound for GLM CMDP

We construct a family of hard instances for the GLM-CMDP problem by building up on the construction of (Osband & Van Roy, 2016) and (Jaksch et al., 2010) and show the following lower bound[2]:

**Theorem 6.1.** *For any algorithm* **A***, there exists a set of values for* $\{S, A, H\}$*, CMDP's with $S$ states, $A$ actions, horizon $H$ and $K \geq dSA$ for logit and linear combination case, such that the expected regret of* **A** *(for any sequence of initial states $\mathcal{S}^K$) after $K$ episodes is $\Omega(H\sqrt{dSAK})$.*

## 7. Related Work

**Contextual MDP** Online learning in a sequence of MDPs indexed by contextual information has been previously studied by (Abbasi-Yadkori & Neu, 2014), (Hallak et al., 2015), (Modi et al., 2018) and (Dann et al., 2018). (Abbasi-Yadkori & Neu, 2014) considered an online learning scenario where the values $p_k(s'|s, a)$ is parameterized by a GLM. The authors give a no-regret algorithm which uses confidence sets based on (Abbasi-Yadkori et al., 2012). However, their next state distributions are not normalized and as such, their model considers invalid probability distributions. (Modi et al., 2018) formalized the CMDP setting and considered a smoothly parameterized CMDP and the linear combination considered in this paper. Their algorithms and analysis deal with the PAC setting and are not directly comparable. (Dann et al., 2018) studied the problem of providing policy certificates and proposed an improved analysis for the linear combination case. They also consider a per next-state linear model which introduces an extra $\sqrt{S}$ factor.

**(Generalized) linear models and partial feedback** Our reward model is based on (stochastic) linear bandits which has seen a vast number of studies since (Abe et al., 2003). Our work leverages ideas from (Abbasi-Yadkori et al., 2011) for both the reward model and intermediate results for the GLM case. Extending the linear bandit problem, (Filippi et al., 2010) first proposed the GLM bandit problem and provided a no-regret algorithm in the OFU paradigm. This paper builds up on the ideas from (Zhang et al., 2016) and (Jun et al., 2017) who also studied the logistic bandit and GLM Bernoulli bandit case. We provide the extension of both methodologies to a generic multinomial GLM setting. Consequently, our bounds also show a dependence on the strong convexity parameter $\frac{1}{\alpha}$ which was recently shown to be unavoidable by (Foster et al., 2018) for proper learning in online logistic regression. Further, (Li et al., 2017) have recently proposed an optimal regret bound for the GLM Bernoulli bandit problem by using phases for removing the statistical dependence between parameter estimates and covariates. However, the proposed algorithm is inefficient and obtaining such an optimal algorithm is still an open problem.

**Regret analysis in MDPs** (Auer & Ortner, 2007) first proposed a no-regret online learning algorithm for average reward infinite horizon MDPs, and the problem has been extensively studied afterwards. More recently, there has been an increased focus on fixed horizon problems where the gap between the upper and lower bounds has been effectively closed. (Azar et al., 2017) and (Dann et al., 2018), both provide optimal regret guarantees ($\tilde{\mathcal{O}}(\sqrt{HSAT})$) for repeated interaction with a finite episodic MDP by using empirical Bernstein based bonus and a fine-grained analysis. Both these methods do not directly carry over to our setting and therefore, improving the dependence on $S, H$ is an interesting problem for future work which would require a novel approach.

## 8. Conclusion and Future Work

In this paper, we have proposed a no-regret algorithm for contextual MDPs which are parameterized by generalized linear models. We provide an ONS based and online regret to confidence set conversion based method for constructing confidence sets. One potential direction for future work is to aim for an efficient and sparsity aware regret bound. In addition, a very important direction to consider is to give a mistake bound simultaneously with the regret guarantee. Further, as a byproduct of our online updates, we don't need to store the contexts thereby adding a layer of privacy. It would be interesting to pursue an algorithm with a corresponding differential privacy style guarantee for the setting. Lastly, closing the gap between the lower and upper bounds is desirable.

---

[2]The proof is deferred to the appendix due to space constraints.

## Acknowledgements

We thank Satinder Singh for helpful discussions and suggestions. AM also thanks Alekh Agarwal for helpful discussions and references. This work was supported in part by a grant from the Open Philanthropy Project to the Center for Human-Compatible AI, and in part by NSF grant CAREER IIS-1452099. AT would like to acknowledge the support of a Sloan Research Fellowship. Any opinions, findings, conclusions, or recommendations expressed here are those of the authors and do not necessarily reflect the views of the sponsors.

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

## A. Proof of the Mistake Bound

In order to prove the mistake bound, we need to bound the number of episodes where the policy's value is more than $\epsilon$-suboptimal. We start with inequality 16 in the main text:

$$V_{k,1}^*(s) - V_{k,1}^{\pi_k}(s)$$
$$\leq \sum_{h,s,a} \mathbb{P}_k[s_h, a_h = s, a | s_{k,1}](2\varphi(s,a) \wedge V_h^{\max})$$

We note that if $\varphi(s,a) \leq \frac{\epsilon}{2H}$ for all $(s,a)$, then we have

$$
\begin{aligned}
V^*_{k,1}(s) - V^{\pi_k}_{k,1}(s) &\leq \sum_{h,s,a} \mathbb{P}_k[s_h, a_h = s, a | s_{k,1}] \frac{\epsilon}{H} \\
&\leq \epsilon
\end{aligned}
$$

In order to satisfy the constraint, we bound each error term as: $\xi^{(p)} \leq \frac{\epsilon}{4H^2}$ and $\xi^{(r)} \leq \frac{\epsilon}{4H}$.

We bound the number of episodes where this constraint is violated. For simplicity, we consider that the rewards are known and only consider the transition probabilities in the analysis:

$$
\begin{aligned}
&\sum_{k \in [K]} \mathbb{I}\left[\exists(s,a) \text{ s.t. } \xi^{(p)}_{k,sa} \geq \frac{\epsilon}{4H^2}\right] \\
&\leq \sum_{k \in [K]} \sum_{s,a} \mathbb{I}[\beta\sqrt{S}\sqrt{\gamma_{k,sa}} \|x_k\|_{Z^{-1}_{k,sa}} \geq \frac{\epsilon}{4H^2}] \\
&\leq \sum_{k \in [K]} \sum_{s,a} \frac{16\beta^2 SH^4 \gamma_{k,sa}}{\epsilon^2} \|x_k\|^2_{Z^{-1}_{k,sa}} \\
&\leq \frac{16\beta^2 SH^4 \gamma_T}{\epsilon^2} \sum_{k \in [K]} \sum_{s,a} \|x_k\|^2_{Z^{-1}_{k,sa}} \\
&\leq \frac{16\beta^2 H^4 \gamma_T}{\epsilon^2} \sum_{s,a} \sum_{k \in [K]} \|x_k\|^2_{Z^{-1}_{k,sa}}
\end{aligned}
$$

where in the intermediate steps, we have used the nature of the indicator function and the fact that minimum is upper bounded by the average. For bounding the inner term, we first consider:

$$
\begin{aligned}
\|x_k\|^2_{Z^{-1}_{k+1}} &= x_k^\top (Z_k + N_k x_k x_k^\top) x_k \\
&= x_k^\top Z_k x_k - \frac{N_k x_k^\top Z_k^{-1} x_k x_k^\top Z_k^{-1} x_k}{1 + N_k x_k^\top Z_k^{-1} x_k} \\
&= \|x_k\|^2_{Z^{-1}_k} - \frac{N_k \|x_k\|^4_{Z^{-1}_k}}{1 + N_k \|x_k\|^2_{Z^{-1}_k}}
\end{aligned}
$$

With this setup, we get:

$$
\begin{aligned}
\|x_k\|^2_{Z^{-1}_k} &= \frac{\|x_k\|^2_{Z^{-1}_{k+1}}}{1 - N_k \|x_k\|^2_{Z^{-1}_{k+1}}} \\
&\leq \frac{\lambda + H}{\lambda} \|x_k\|^2_{Z^{-1}_{k+1}}
\end{aligned}
$$

Therefore, the inner summation can be instead bounded as $\frac{\lambda+H}{\lambda} \sum_{k \in [K]} \|x_k\|^2_{Z^{-1}_{k+1}}$. Using Lemma 11 from (Hazan et al., 2007), we can bound the sum as $d\log\left(\frac{R^2 T}{\lambda} + 1\right)$. Combining all these bounds, we get:

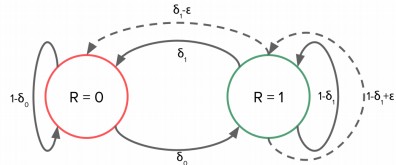

Figure 1. Hard 2-state MDP (Osband & Van Roy, 2016)

$$
\begin{aligned}
&\sum_{k \in [K]} \mathbb{I}\left[\exists(s,a) \text{ s.t. } \xi^{(p)}_{k,sa} \geq \frac{\epsilon}{4H^2}\right] \\
&\leq \frac{16(\lambda+H)\beta^2 dS^2 AH^4 \gamma_T}{\lambda\epsilon^2} \log\left(\frac{R^2 T}{\lambda} + 1\right)
\end{aligned}
$$

Noting that $\gamma_T = \mathcal{O}(\frac{d\log^2 T}{\alpha} + S)$, we get the final mistake bound as:

$$
\mathcal{O}\left(\frac{dS^2 AH^5 \log T}{\epsilon^2}\left(\frac{d\log^2 T}{\alpha} + S\right)\right)
$$

ignoring $\mathcal{O}(\text{poly}(\log\log T))$ terms.

## B. Proof of the Lower Bound

*Proof.* We start with the lower bound from (Jaksch et al., 2010) adapted to the episodic setting.

**Theorem B.1** ((Jaksch et al., 2010), Thm. 5). *For any algorithm* $\mathbf{A}'$, *there exists a set of values for* $\{S, A, H\}$, *an MDP $M$ with $S$ states, $A$ actions, and horizon $H$, such that for $K \geq dSA$, the expected regret of $\mathbf{A}$ after $K$ episodes is:*

$$
\mathbb{E}[R(K; \mathbf{A}', s, M)] = \Omega(H\sqrt{SAK})
$$

The lower bound construction is obtained by concatenating $\lceil S/2 \rceil$-copies of a bandit-like 2-state MDP as shown in figure 1[3]. Essentially, state 1 is a rewarding state and all but one action take the agent to state 0 with probability $\delta_1$. The remaining optimal action transits to state 0 with probability $\delta_1 - \epsilon$. This makes the construction similar to a hard Bernoulli multi-armed bandit instance which leads to the lower bound. Now, we will construct a set of such hard instances with the logit link function for transition probabilities. A similar construction for the linear combination case is discussed in Appendix B. Since, the number of next states is 2, we use a GLM with parameter vector $w^*$ of shape $1 \times d$. Thus, for any context $x$, the next state probabilities are given as:

$$
p(1|1, a; x) = \frac{\exp(w_a^* x)}{1 + \exp(w_a^* x)} = \phi(w_a^* x)
$$

---

[3]The two state MDP is built using $A/2$ actions with the rest used for concatenation. We ignore this as it only leads to a difference in constants.

If $w_a^* x = 0$, the value turns out to be $\frac{1}{2}$ which we choose as $\delta_1 - \epsilon$. For making the probability $\delta_1 = \frac{1}{2} + \epsilon$, we need to have $w_a^* x = \phi^{-1}(\delta_1) = c^*$. We consider the case where for each index $i$, all but one action has $w_a^*[i] = 0$ and one action $a_i^*$ has $w_{a*}^*[i] = c^*$. The sequence of contexts given to the algorithm comprises of $K/d$ indicator vectors with 1 at only one index. Therefore, for each episode $k$, we get an MDP with $p_k(0|1, a_{k\%d}^*) = 1/2$ for one optimal action and $1/2$ for all other actions. Therefore, this is a hard instance as shown in figure 1. The agent interacts with each such MDP $K_i \approx K/d$ times. Further, these MDPs are decoupled as the context vectors are non-overlapping. Therefore, we have:

$$\mathbb{E}[R(K; \mathbf{A}, M_{1:K}, s_{1:K})]$$
$$= \sum_{i=1}^{d} \mathbb{E}[R(K_i; \mathbf{A}, M_{1:K}, s_{1:K})]$$
$$\geq \sum_{i=1}^{d} cH\sqrt{SAK/d} = cH\sqrt{dSAK}$$

$\square$

**Linear combination case**   Similar to the logit case, we need to construct the sequence of hard instances in the linear combination case. It turns out that a similar construction works. Note that, in the linear combination case, each parameter vector $w_a^*$ now directly contains the probability of moving to the rewarding state. In other words, each index of this vector $w_a^*[i]$ corresponds to the next state visitation probability for the base MDP $M_i$. Therefore, for each index, we again set one action's value to $\frac{1}{2} + \epsilon$ and all others to 0. This maintains the independence argument and using indicator vectors as contexts, we get the same sequence of MDPs. The same lower bound can therefore be obtained for the linear combination case.

## C. Omitted Proofs from Section 5

**Theorem C.1** (Multinomial GLM Online-to-confidence set conversion). *Suppose losses $l_i$ are $\alpha$-strongly convex. If there exists an online learning oracle which takes in the sequence $\{x_i, y_i\}_{i=1}^{t}$, and produces outputs $\{W_i\}_{i=1}^{t}$ with bounded regret for all $W \in \mathcal{W}$ and $t \geq 1$:*

$$\sum_{i=1}^{t} l_i(W_i) - l_i(W) \leq B_t,$$

*then with the centers $\overline{W}_t$ defined above, we have, with probability at least $1 - \delta$, for all $t \geq 1$, we have*

$$\|W^* - \overline{W}_t\|_{Z_{t+1}}^2 \leq \gamma_t$$

*where $\gamma_t := \gamma_t'(B_t) + \lambda B^2 S - (\|C_t\|_F^2 - \langle \overline{W}_t, X_t^\top C_t \rangle)$ with*

$$\gamma_t'(B_t) := 1 + \frac{4}{\alpha} B_t + \frac{8}{\alpha^2} \log\left(\frac{1}{\delta}\sqrt{4 + \frac{8B_t}{\alpha} + \frac{16}{\alpha^4 \delta^2}}\right).$$

*Proof.* Using the strong convexity of the losses $l_i$, we again have:

$$l_i(W_i) - l_i(W^*)$$
$$\geq \langle \nabla l_i(W^*), W^* - W_i \rangle + \frac{\alpha}{2}\|W^* x_i - W_i x_i\|_2^2$$

Summing this for $i = 1$ to $t$ and substituting the regret bound $B_t$, we get

$$\sum_{i=1}^{t} \|W^* x_i - W_i x_i\|_2^2$$
$$\leq \frac{2}{\alpha} B_t + \frac{2}{\alpha} \sum_{i=1}^{t} \langle p_t - y_t, W^* x_i - W_i x_i \rangle \quad (17)$$

Now, we focus on bounding the second term in the rhs. We note that for any $z \in \mathbb{R}^S$, we have

$$\langle p_t - y_t, z \rangle \leq \|p_t - y_t\|_2 \|z\|_2 \leq 2\|z\|_2$$

In addition, $\langle \eta_t, z \rangle := \langle p_t - y_t, z \rangle$ is a martingale with respect to the filtration $\mathcal{F}_t := \sigma(x_1, y_1, \ldots, x_{t-1}, y_{t-1}, x_t)$. This shows that

$$\mathbb{E}[D_t^\lambda | \mathcal{F}_t] = \mathbb{E}[\exp(\lambda\langle \eta_t, z \rangle - \tfrac{1}{2}\lambda^2 \|z\|_2^2)|\mathcal{F}_t] \leq 1$$

We can substitute $z_t = W^* x_t - W_t x_t$ which is $\mathcal{F}_t$ measurable. Now, using $S_t = \sum_{i=1}^{t} \langle \eta_i, z_i \rangle$, we can show that $M_t^\lambda = \exp\left(4\lambda S_t - \frac{1}{2}\lambda^2 \sum_{i=1}^{t} \|z_i\|_2^2\right)$ is a $\mathcal{F}_{t+1}$-adapted supermartingale. Using the same analysis as in (Abbasi-Yadkori et al., 2012), we get the following result:

**Corollary C.2** (Corollary 8, (Abbasi-Yadkori et al., 2012))**.** *With probability at least $1 - \delta$, for all $t > 0$, we have*

$$\sum_{i=1}^{t} \langle \eta_i, z_i \rangle$$
$$\leq \sqrt{2\left(1 + \sum_{i=1}^{t} \|z_i\|_2^2\right) \ln\left(\frac{1}{\delta}\sqrt{\left(1 + \sum_{i=1}^{t} \|z_i\|_2^2\right)}\right)}$$

Substituting this in ineq. 17, we get

$$\sum_{i=1}^{t} \|z_i\|_2^2 - \frac{2}{\alpha} B_t$$
$$\leq \frac{2}{\alpha}\sqrt{2\left(1 + \sum_{i=1}^{t} \|z_i\|_2^2\right) \ln\left(\frac{1}{\delta}\sqrt{\left(1 + \sum_{i=1}^{t} \|z_i\|_2^2\right)}\right)}$$

We now use Lemma 2 from (Jun et al., 2017), to obtain a simplified bound:

**Lemma C.3** (Lemma 2, (Jun et al., 2017)). *For $\delta \in (0, 1)$, $a \geq 0, f \geq 0, q \geq 1, q^2 \leq a + fq\sqrt{\log \frac{q}{\delta}}$ implies*

$$q^2 \leq 2a + f^2 \log \Big( \frac{\sqrt{4a + f^4/(4\delta^2)}}{\delta} \Big)$$

With $q := \sqrt{1 + \sum_{i=1}^{t} \|z_i\|_2^2}$, $a := 1 + \frac{2}{\alpha} B_t$ and $f = \frac{2\sqrt{2}}{\alpha}$, we now have:

$$\sum_{i=1}^{t} \|W^* x_i - W_i x_i\|_2^2 \leq \gamma_t' \tag{18}$$

with $\gamma_t' := 1 + \frac{4}{\alpha} B_t + \frac{8}{\alpha^2} \log \Big( \frac{1}{\delta} \sqrt{4 + \frac{8B_t}{\alpha} + \frac{16}{\alpha^4 \delta^2}} \Big)$.

We can rewrite ineq. 18 as

$$\|X_t W^{*\top} - C_t\|_F^2 \leq \gamma_t' \tag{19}$$

If we center this quadratic form around

$$\overline{W}_t := \arg\min_{W} \|X_t W^\top - C_t\|_F^2 + \lambda \|W\|_F^2 = Z_{t+1}^{-1} X_t^\top C_t$$

we can rewrite the set as:

$$\|W^* - \overline{W}_t\|_{Z_{t+1}}^2$$
$$\leq \lambda B^2 S + \gamma_t' - (\|\overline{W}_t\|_F^2 + \|X_t \overline{W}_t^\top - C_t\|_F^2)$$

Simplifying the expression on the rhs gives the stated result. $\square$