# OpenReview forum: "Contextual Markov Decision Processes using Generalized Linear Models"
_ICML.cc/2019/Workshop/RL4RealLife — RL4RealLife 2019_

### Official Review · AnonReviewer2 · 2019-05-24
**The paper generalizes previous contextual MDP models, develops an order-optimal algorithm leveraging known algorithms, and proves its performance bound.  NOt sure how suitable the venue is given the purely theoretical nature of the work.**

**Rating:** 4
**Confidence:** 4

**Review:**

The authors study finite horizon contextual Markov Decision Process (MDP) problems using generalized linear models. In particular, at each episode, a context is given a priori. The transition matrix of the MDP is assumed to follow a generalized linear model for the given context. The model is shown to be general that subsumes previously studied multinomial logit model and linear combination model. Following the popular optimism under uncertainty approach, the authors propose a no-regret algorithm that relies on the construction of confidence sets for MDP parameters at the beginning of each episode. The authors prove the performance bound of the proposed algorithm, and improve the performance of prior bound on multinomial GLM algorithms.

The paper generalizes previous contextual MDP models, develops an order-optimal algorithm leveraging known algorithms, and proves its performance bound.

There is a recent literature on episodic reinforcement learning that should be discussed and compared with in related work.

Given the limited time, I did not go through the analysis in detail. It does look correct in general.

The work is theoretical in nature. I am wondering how such algorithms perform in practice. Also I wonder whether this is the most suitable venue. The audience may not be as excited as some other audience could be.

---

### Official Review · AnonReviewer1 · 2019-05-25
**Interesting problem, approach, and results**

**Rating:** 5
**Confidence:** 4

**Review:**

In this paper, the authors study RL in contextual MDP. This an episodic RL problem where at the beginning of each episode, the agent receives a context for which the MDP of that episode is function off. In this setting, the authors study the case where the transition kernel of each MDP is specified with a generalized linear model of the context. Here the rewards are linear functions of context. Through interaction, the agent learns the reward function as well as the transition kernel, deploys OFU and provide PAC and regret bounds.

Novelty: to the best of my knowledge, this problem is a straight forward extension of prior works, and I do not any paper that has approached this problem before.

The paper is well written. Just emphasize more that it is tabular MDP.
I did not fully check the proof, but it should be pretty standard given the prior works.

Please also cite this paper,
https://arxiv.org/pdf/1502.02259.pdf

---

### Decision · Program_Chairs · 2019-05-28

Accept